# Exploring the Correlation between the PASI and DLQI Scores in Psoriasis Treatment with Topical Ointments Containing *Rosa × damascena* Mill. Extract

**DOI:** 10.3390/ph17081092

**Published:** 2024-08-20

**Authors:** Diana Ioana Gavra, Dóra Kósa, Ágota Pető, Liza Józsa, Zoltán Ujhelyi, Pálma Fehér, Annamária Pallag, Timea Claudia Ghitea, Simona Frățilă, Tünde Jurca, Ildikó Bácskay

**Affiliations:** 1Department of Preclinical Discipline, Faculty of Medicine and Pharmacy, University of Oradea, 1st December Square 10, H-410068 Oradea, Romania; diannagavra@yahoo.com (D.I.G.); annamariapallag@gmail.com (A.P.); timea.ghitea@csud.uoradea.ro (T.C.G.); jurcatunde@yahoo.com (T.J.); 2Department of Pharmaceutical Technology, Faculty of Pharmacy, University of Debrecen, Nagyerdei Körút 98, H-4032 Debrecen, Hungary; kosa.dora@pharm.unideb.hu (D.K.); jozsa.liza@euipar.unideb.hu (L.J.); ujhelyi.zoltan@pharm.unideb.hu (Z.U.); feher.palma@pharm.unideb.hu (P.F.); 3Institute of Healthcare Industry, University of Debrecen, Nagyerdei Körút 98, H-4032 Debrecen, Hungary; 4Department of Psycho-Neuroscience and Recovery, Faculty of Medicine and Pharmacy, University of Oradea, 1st December Square 10, H-410068 Oradea, Romania; simonafra_drm@yahoo.com

**Keywords:** psoriasis, *Rosa × damascena* Mill., PASI score, DLQI score, anti-inflammatory

## Abstract

Psoriasis is a chronic autoimmune skin condition characterized by red, circumscribed, scaly, and erythematous plaques that can cover large skin areas. While conventional treatments such as topical corticosteroids and systemic medications are commonly prescribed, the interest in natural remedies for psoriasis has grown due to concerns about potential side effects and the desire for alternative treatment options. *Rosa × damascena* Mill. is rich in bioactive compounds that possess anti-inflammatory, antioxidant, and antimicrobial properties; these properties make *Rosa × damascena* Mill. a promising candidate for the management of skin disorders such as psoriasis. In our previous studies, we successfully formulated and tested different topical preparations containing *Rosa × damascena* Mill. In this study, we investigated the correlation between the Psoriasis Area and Severity Index (PASI) and Dermatology Life Quality Index (DLQI) scores in psoriasis treatment using the abovementioned creams containing *Rosa × damascena* Mill. extract. Several tests were performed to study the correlation between the PASI and DLQI scores in psoriasis patients. Consequently, we were able to observe an improvement in terms of the area, induration, desquamation, and erythema; such an improvement implicitly produces an improvement in patients’ quality of life. The PASI and DLQI scores showed significant progress between visits. These results confirm *Rosa × damascena* Mill. to be a promising candidate for the topical treatment of psoriatic lesions.

## 1. Introduction

Psoriasis is a chronic inflammatory disorder characterized by its erythematous and scaly skin lesions, which often lead to considerable physical discomfort and emotional distress. While the exact etiology of psoriasis remains incompletely understood, significant advancements in research have shed light on its pathogenesis at the cellular and molecular levels. Psoriasis is fundamentally driven by dysregulation of the immune system, particularly T lymphocytes, dendritic cells, and cytokines [1]. In predisposed individuals, environmental triggers, such as infections or trauma, can activate innate immune pathways, leading to the production of proinflammatory cytokines, including tumor necrosis factor-alpha (TNF-α), interleukin-17 (IL-17), and interleukin-23 (IL-23), which lead to chronic inflammation and aberrant keratinocyte proliferation in the skin. Central to the pathogenesis of psoriasis is the activation of T lymphocytes, which stimulate keratinocytes to proliferate and produce inflammatory mediators [2]. In psoriasis, keratinocytes undergo hyperproliferation and aberrant differentiation, contributing to the thickening of the epidermis and the formation of characteristic psoriatic plaques. Disruption of the normal keratinocyte turnover cycle, known as the epidermal hyperproliferative state, is mediated by cytokines, which stimulate keratinocyte proliferation and inhibit terminal differentiation. Additionally, the altered expression of the genes involved in epidermal barrier function and inflammation further exacerbates keratinocyte abnormalities in psoriasis. Angiogenesis, the formation of new blood vessels, is a hallmark feature of psoriasis and contributes to the characteristic erythema and vascularity of psoriatic plaques [3].

Targeted therapies aimed at modulating the specific molecular pathways involved in psoriasis pathogenesis have revolutionized the treatment landscape, offering new hope for improved disease management and better outcomes for patients affected by this chronic condition. While conventional treatments such as topical corticosteroids and systemic medications are commonly prescribed, the interest in natural remedies for psoriasis has grown due to concerns about potential side effects and the desire for alternative treatment options.

Herbal remedies have been used for centuries in traditional medicines for managing various skin conditions, including psoriasis. Studies have highlighted the potential of herbs such as *Aloe vera* L. [4], *Curcuma longa* L. [3,5,6], and *Avena sativa* L. [7] in reducing the inflammation, itching, and scaling associated with psoriasis. Another natural drug of considerable interest is *Rosa × damascena* Mill. (commonly known as Damask rose), which is rich in bioactive compounds and possesses several relevant properties, such as anti-inflammatory [8], antioxidant [9], and antimicrobial [10,11,12] effects. Given these attributes, it could potentially be a promising candidate for the management of inflammatory skin disorders, including psoriasis. The antioxidant compounds, including flavonoids, phenols, and essential oils, of *Rosa × damascena* Mill. can neutralize the increased oxidative stress that is a significant factor in the pathogenesis of psoriasis [13]. In our previous study, we demonstrated that an ointment containing *Rosa × damascena* Mill. extract exhibits significant antioxidant effects, potentially beneficial for treating psoriasis [14]. Building upon these findings, the current clinical study investigates the efficacy of this ointment in a broader patient population.

Plant extracts are generally considered less harmful than synthetic drugs, making them suitable for prolonged use in managing systemic conditions such as psoriasis. Our previous research indicated that *Rosa × damascena* extract not only inhibits proinflammatory cytokines but also enhances skin hydration and barrier function [14]. While synthetic drugs like TNF inhibitors, IL-17 inhibitors, and IL-23 inhibitors offer rapid relief by targeting specific immune components, they can pose significant risks of side effects, including increased susceptibility to infections and organ toxicity [15,16,17,18,19]. Conversely, natural extracts, though slower in showing effects, often provide a gentler and safer approach, making them ideal for long-term management and for patients seeking alternative therapies [19,20]. *Rosa × damascena* Mill. stands out due to its comprehensive anti-inflammatory, antioxidant, and skin barrier-improving properties. Its gentle action and multi-faceted benefits make it a particularly advantageous choice for treating skin lesions in psoriasis. Compared to some other plant extracts that can be harsh or cause irritation, *Rosa × damascena* Mill. is suitable for sensitive skin and long-term use without significant side effects [14,21,22,23,24].

Our prior in vitro studies have explored its phytochemical profile and biological activity [14,21], but the beneficial in vivo effects of this extract have only been demonstrated in a limited patient sample [14]. This article aims to expand on this research by conducting a clinical study to evaluate the efficacy of *Rosa × damascena* Mill. in a larger cohort of psoriasis patients and to provide a scientific basis for its use as a topical therapy.

The present study focuses on the comparison of the Psoriasis Area and Severity Index (PASI) and Dermatology Life Quality Index (DLQI) scores in the treatment of psoriasis using topical ointments containing *Rosa × damascena* Mill. extract versus a placebo ointment containing only the vehicle. Our research evaluates the impact of these extracts on the PASI and DLQI scores, aiming to demonstrate both clinical and quality-of-life improvements in patients. Through this focused approach, we seek to substantiate the potential of *Rosa × damascena* Mill. as a viable alternative or adjunct to conventional psoriasis treatments.

When conducting a clinical study on topical treatments for psoriasis, careful patient selection is crucial to ensure the validity and generalizability of the results; this involves considering the disease severity, excluding confounding factors, ensuring a representative sample, addressing ethical considerations, and considering practical aspects such as adherence and follow-up. Proper patient selection helps to ensure that the study findings are reliable and applicable to the broader psoriasis patient population [25].

Effective management of psoriasis involves regular monitoring to assess the disease severity, treatment efficacy, and patient quality of life [26,27]. To evaluate the effectiveness of the rose extract in patients with psoriasis, the PASI and DLQI scores were utilized. The PASI is the most widely used tool for measuring the severity and extent of psoriasis. The PASI score considers the degree of erythema (redness), induration (thickness), desquamation (scaling), and the percentage of body surface area affected. Each of these parameters is assessed for different body regions: the head, upper limbs, trunk, and lower limbs [28]. The DLQI is a validated questionnaire used to assess the impact of psoriasis on the quality of life. The DLQI consists of 10 questions covering symptoms and feelings, daily activities, leisure, work and school, personal relationships, and treatment. Each question is scored from 0 to 3, resulting in a total score ranging from 0 to 30. Higher scores indicate a greater impact on the daily functioning and well-being of the patient [29,30].

## 2. Results

A statistical analysis of the two groups that took part in the study was conducted, as shown in the figure and table below. Group 1 consisted of patients with psoriasis who received topical treatment with *Rosa × damascena* Mill. extract, and the second group was the placebo group, which comprised the participants who received a placebo treatment. These two groups were observed over a period of 6 weeks. During the study, the PASI score parameters and the DLQI score were analyzed during three visits, as can be seen in Figure 1 and Table 1. The area did not differ significantly between visit 1 and visit 2, between visit 1 and visit 3, or between visit 2 and visit 3. The erythema differed significantly between visit 1 and visit 2 and between visit 1 and visit 3, as well as between visit 2 and visit 3. The desquamation also differed significantly between visit 1 and visit 2 and between visit 1 and visit 3, as well as between visit 2 and visit 3. The induration also differed significantly between visit 1 and visit 2 and between visit 1 and visit 3, as well as between visit 2 and visit 3. The PASI and DLQI scores showed a significant improvement between the three visits for Group 1. The area differed significantly between Group 1 and Group 2 at visits 1, 2, and 3, but the greatest significance could be observed at visit 3 (mean = 0.5515 > 0.2151 > 0.2169), with F = 16.369 and Sig = 0.001. For the erythema parameter, when comparing the two groups, the highest significance could be observed at visit 3 (mean = 0.9421). Regarding induration, a significant difference between the group of patients with psoriasis and the placebo group could be observed at the third visit (mean = 0.6471), resulting in *p* < 0.05.

An important assessment of the effectiveness of the treatment can also be performed by evaluating the degree of desquamation. At each visit, this parameter was also evaluated; when comparing the two groups, the desquamation was significantly different at the second and third visits (mean = 0.1875, respectively, 0.4779), with *p* < 0.05.

Moreover, considering the parameters mentioned above and evaluating the patients’ quality of life, at each visit the PASI and DLQI scores were calculated; these scores also differed significantly at visit 3 (mean = 1.4456, respectively, 1.6029 for the DLQI score).

Table 2 and Figure 2 show the risk of an increase in the six parameters for the group of patients who used only the placebo ointment compared to the patients in Group 1 who had topical treatment with *Rosa × damascena* extract. In Figure 2, we can see that the risk for the area is 86.4% for the placebo group patients compared to the Group 1 patients who used the ointment with *Rosa × damascena* Mill. extract. From the value of the Pearson correlation (r = 0.936), a statistically significant positive correlation at *p* < 0.01 can be observed for erythema. The risks of induration and desquamation for the two groups of patients, when considering the value of the Pearson correlation, *p* < 0.01, are 95.2% and 92.9%, respectively. By calculating the PASI and DLQI scores for the two groups studied, we can state that Group 2 has the highest risk (97.6%) of an increase in the PASI score; thus, the quality of life of these patients is significantly affected and there is a risk of 95.6% for this group of patients.

## 3. Discussion

While conventional treatments such as topical corticosteroids and systemic medications are effective for many psoriasis patients, there is growing interest in the use of topical creams prepared with natural extracts as alternative or adjunctive therapies [31]. In this study, we explored the correlation between the PASI and DLQI scores in psoriasis treatment by studying two groups of patients. Group 1 was treated with topical ointments containing *Rosa × damascena* Mill. extract, and Group 2 was used as a placebo group that received an ointment with the same composition as that of the first group, except for the extract with *Rosa × damascena* Mill.

Because it is rich in bioactive compounds, such as flavonoids, phenolics, and essential oils, *Rosa × damascena* Mill. possesses anti-inflammatory, antioxidant, and antimicrobial properties that make it a promising candidate for managing skin disorders such as psoriasis. The phenolic compounds found in *Rosa × damascena* Mill. include quercetin, which suppresses the activation of the NF-κB pathway, scavenges free radicals, and reduces oxidative stress; kaempferol, which inhibits the expression of cyclooxygenase-2 (COX-2) and nitric oxide synthase (iNOS) and modulates immune responses by affecting T-cell proliferation and cytokine production; gallic acid, which inhibits the production of inflammatory cytokines and enzymes, such as COX-2 and lipoxygenase (LOX), and has strong antioxidant capabilities; tannins, which reduce the infiltration of inflammatory cells into the skin and help to contract and tighten tissues, which can reduce the swelling and erythema associated with psoriasis; and anthocyanins, which reduce oxidative stress by scavenging free radicals and enhance the skin’s barrier function, which can help protect against environmental triggers that may worsen psoriasis. These compounds play significant roles in *Rosa × damascena* Mill.’s anti-inflammatory effects on psoriasis [22,23,24].

To the best of our knowledge, specific studies investigating the use of *Rosa × damascena* in the treatment of psoriasis are limited. However, its significant anti-inflammatory and antioxidant effects were proved. According to the in vivo study by Hajhashemi et al., the extract of *Rosa × damascena* has significant anti-inflammatory activity when investigated by a carrageenan-induced edema test in rats [8].

Psoriasis is also characterized by oxidative stress, where an imbalance between antioxidants and reactive oxygen species (ROS) leads to cellular damage and inflammation. *Rosa × damascena* Mill. is known for its high antioxidant content, which confers protective effects against oxidative stress-induced skin damage. Researchers have highlighted the antioxidant potential of *Rosa × damascena* Mill. extract in scavenging free radicals and reducing oxidative stress markers in skin cells [21,22]. Kumar et al. demonstrated the presence of phenolic compounds in the ethanolic extract of *Rosa × damascena*. They assessed the antioxidant activity of this extract in comparison to the standard antioxidant L-ascorbic acid using the 1,1-diphenyl-2-picrylhydrazyl (DPPH) free-radical method. The study also revealed that *Rosa × damascena* exhibits high antioxidant activity [32]. Sedighi et al. demonstrated the antioxidant activity of a 70% hydro-alcoholic extract of Damask rose using the ferric thiocyanate method, showing an activity level equivalent to 78% of rutin, a standard flavonoid compound [33]. In another study conducted by Dadkhah and his research group, it was found that the treatments of rats with *Rosa × damascena* Mill. essential oils diminished the levels of oxidative stress [34]. By supporting the antioxidant defense mechanisms of the skin, *Rosa × damascena* Mill. may help to mitigate oxidative damage and promote skin healing in psoriasis patients.

It was also described that *Rosa × damascena* Mill. extract exhibits potent anti-inflammatory effects by inhibiting key proinflammatory cytokines in epidermal keratinocytes, including TNF-α, IL-6, IL-17, and IL-23. Its antioxidant properties and ability to inhibit the NF-κB pathway further contribute to its therapeutic potential in psoriasis treatment, making it a valuable alternative or adjunct to conventional synthetic drugs [21,22]. According to a recent study conducted by Wedler et al., *Rosa × damascena* extract markedly modified inflammatory target gene expression in vitro, and therefore, could be further developed as alternative treatment of acute and chronic inflammation [35].

These studies have shown that the application of *Rosa × damascena* Mill. extract can suppress proinflammatory cytokines and inhibit the inflammatory pathways implicated in psoriasis pathogenesis. By mitigating inflammation, *Rosa × damascena* Mill. may help to alleviate symptoms such as redness, swelling, and itching, which are associated with psoriatic lesions.

The Psoriasis Area and Severity Index (PASI) and Dermatology Life Quality Index (DLQI) are two widely used tools for assessing the disease severity and its impact on patients’ daily functioning and well-being. The PASI score provides an objective and standardized method for assessing the severity of psoriasis lesions. By quantifying the extent and intensity of the erythema, induration, and desquamation across different body regions, the PASI score offered us a comprehensive evaluation of the disease severity [29]. During our study, the visits of the patients at a predetermined time permitted us to monitor changes in their psoriasis lesions over time and to objectively assess the treatment response. The changes in the PASI scores before and after treatment initiation served as a reliable indicator of the treatment efficacy. The reductions in the PASI scores signified improvements in the disease severity and lesion clearance, reflecting the therapeutic effect of the treatment. While the PASI score provides valuable information about the disease severity from a clinician’s perspective, it is essential to consider the patients’ perspectives and subjective experiences when evaluating treatment efficacy. Patient-reported outcomes, such as quality of life assessments and satisfaction with treatment, complement the PASI score and provide insights into the overall impact of psoriasis on patients’ lives. Integrating patient-centered measures with objective clinical assessments ensures a comprehensive evaluation of treatment efficacy and patient well-being [36].

On the other hand, the DLQI is a subjective assessment tool that captures the impact of psoriasis on various aspects of patients’ lives, including symptoms and feelings, daily activities, leisure, work, personal relationships, and treatment-related concerns. By capturing the physical, emotional, and social dimensions of the disease, the DLQI offers a comprehensive perspective on treatment efficacy beyond traditional clinical measures such as disease severity scores. One of the strengths of the DLQI is its patient-centered nature, allowing individuals to express their subjective experiences and perceptions of their skin condition and its impact on their quality of life. Changes in the DLQI scores can reflect not only improvements in physical symptoms but also enhancements in psychological well-being, social functioning, and overall satisfaction with treatment outcomes [30,31].

As we could observe in our patients, the reductions in the DLQI scores following treatment initiation signified improvements in their quality of life and functional status, indicating a positive response to therapy. By regularly assessing the DLQI scores at predefined intervals, clinicians can identify trends in patient-reported outcomes, assess treatment adherence and tolerability, and modify therapeutic regimens as needed to optimize outcomes and enhance patient satisfaction.

Several studies have investigated the correlation between the PASI and DLQI scores in psoriasis patients undergoing treatment with topical creams prepared with natural extracts. For example, in a recent clinical study, Maul et al. described that an increment in the PASI score affected the quality of life more severely in patients whose quality of life is already highly affected [37]. In our study, we were able to observe, as shown above, an improvement in terms of the area, induration, desquamation, and erythema; such an improvement implicitly produces an improvement in patients’ quality of life, reflecting the close relationship between disease activity and its impact on patients’ well-being. Clinical trials evaluating the efficacy of these natural extracts in topical formulations have reported significant reductions in the PASI scores, indicating improvements in the disease severity and lesion clearance, but there are not many studies reported in the literature that use *Rosa × damascena* Mill. extract [38,39,40]. That is why our work could potentially fill a significant gap in the literature and contribute valuable insights into the efficacy of *Rosa × damascena* for psoriasis treatment.

Studies have reported improvements in the DLQI score following treatment with natural extract-based creams, reflecting improvements in patients’ physical symptoms, psychological well-being, social functioning, and treatment satisfaction. These findings highlight the benefits of natural extract-based therapies beyond the objective measures of disease severity [38,39,40,41].

In this study, we can observe a significant improvement in the affected area, erythema, induration, and desquamation, as well as the DLQI score in Group 1 compared to Group 2 (Table 1). The slight improvement observed in the placebo group was probably due to the emollient effect of the ointment.

By evaluating the results obtained in our study, we can conclude that the best results were obtained in the patients who used the *Rosa × damascena* Mill. extract ointment; the placebo effect in the Group 2 patients in the short term was approximately 50%, but in the long term, this condition requires drug treatment that can be completed with an adjuvant treatment such as *Rosa × damascena* Mill. extract ointment.

## 4. Materials and Methods

This clinical study consisted of patients with a dermatological diagnostic of psoriasis established by a qualified dermatologist. The diagnosis was based on the characteristic signs and symptoms, such as erythematous plaques with silvery scales, nail changes, and the presence of psoriatic lesions in typical locations.

A randomized prospective study was carried out on 126 patients with psoriasis. Patients were selected from a private dermatological office and from a nutrition and dietetics medical office, both from Oradea, Bihor County, Romania, during January to July 2022.

Collecting demographic information, such as age, gender, ethnicity, occupation, lifestyle factors, and medical history, helped us to characterize the study population and identify potential confounders or effect modifiers. Additionally, assessing baseline characteristics related to psoriasis, such as the disease duration, previous treatments, and family history, provided valuable insights into the patient profiles and treatment response predictors. The inclusion criteria comprised individuals aged over 18 years and those diagnosed with a mild form of psoriasis, characterized by a body surface area involvement of less than 10%. The exclusion criteria included pregnancy, lactation, recent use (within the past 3 months) or concurrent use of other topical or systemic medications for psoriasis, the presence of comorbidities that could potentially confound the study results, and a history of hypersensitivity to the medications or components utilized in the study. Clear inclusion and exclusion criteria helped us to ensure that the selected patients were representative and that confounding factors were minimized.

For each patient studied, psoriasis lesions with a similar size and the same clinical severity were selected. Forty-eight patients were selected who met the criteria for inclusion and completed the informed consent. The patients were randomly distributed in two groups, with an equal number of members. Group 1 received a predetermined amount (2.50 g) of topical treatment containing lyophilized extract of *Rosa × damascena* Mill. Group 2 received placebo ointment (2.50 g) containing only the vehicle included in the active ointment. Each topical treatment was applied twice daily (once in the morning and once in the evening, on psoriatic lesions) for 6 weeks. Patients were evaluated visually at the beginning, after three weeks, and at the end of the study. The ointments were formulated in the laboratory of pharmaceutical technology of the Faculty of Pharmacy, Debrecen, Hungary, following the rules of good manufacturing practice. The formulations were analyzed in terms of their physicochemical, microbiological and stability properties.

The evolution of the disease under the topical treatment applied was monitored using the Psoriasis Area and Severity Index (PASI) and the Disease Linked Quality of Life (DLQI) scores. To calculate the PASI score, psoriasis was assessed in four body regions: head, upper limbs, trunk, and lower limbs. For each region, the erythema (redness), induration (thickness), and desquamation (scaling) were evaluated and scored on a scale from 0 to 4 (0 = none, 1 = mild, 2 = moderate, 3 = severe, 4 = very severe). These parameters were scaled based on visual inspections. For the determination of the PASI score, an online PASI calculator was used [42].

The percentage of the body area affected was also scored from 0 to 6, where 0 represents no involvement, 1 = <10%, 2 = 10–29%, 3 = 30–49%, 4 = 50–69%, 5 = 70–89%, and 6 = 90–100% [43].

The DLQI scores were determined based on a questionnaire containing ten questions related to the patient’s quality of life, with the focus on the impact of the dermatological condition on various aspects of daily living. Our questions focused on the impact of physical symptoms and emotional distress caused by psoriasis, the effect of psoriasis on daily activities such as shopping or dressing, its impact on leisure activities including sports and hobbies, its effect on work or school activities, the impact on relationships with family and friends, and the burden of treatment on the life of the patient. Each question was scored on a scale from 0 to 3 (0 = not at all, 1 = a little, 2 = a lot, 3 = very much), and the scores for all ten questions were summed to obtain the total DLQI score, which ranged from 0 to 30 (0–1 = no effect on the patient’s quality of life, 2–5 = small effect, 6–10 = moderate effect, 11–20 = very large effect, 21–30 = extremely large effect).

Clinical photos were taken and the PASI and DLQI scores were calculated at the initial visit and at two follow-up visits, 3 weeks apart. The study was carried out with the consent of the Ethics Commission CEMF/03 from 31 October 2021.

Statistical analysis was performed using the ANOVA program. Participation in the study was voluntary. Informed consent was obtained from each patient included in the study and patient confidentiality was ensured.

## 5. Conclusions

In conclusion, ointments prepared with natural extracts offer a promising approach for the alternative or adjunctive treatment of psoriasis, with potential benefits in terms of a reduction in the PASI scores and improvement of the quality of life in affected patients. The correlation between the PASI and DLQI scores accentuates the importance of addressing both the disease severity and its impact on patients’ daily functioning and well-being.

*Rosa × damascena* Mill. emerges as a botanical gem in the realm of natural remedies for psoriasis. Its multifaceted pharmacological profile, encompassing anti-inflammatory, antioxidant, and antimicrobial properties, positions it as a promising candidate for the topical treatment of psoriatic lesions. Although the pilot studies provide encouraging results, further clinical trials are required to reveal the true potential of *Rosa × damascena* Mill. as a psoriasis therapy. Ultimately, recognizing the holistic properties of this plant could offer hope for the generation of less harmful but still powerful treatments to diminish the burden of psoriasis in patients’ lives.

## Figures and Tables

**Figure 1 pharmaceuticals-17-01092-f001:**
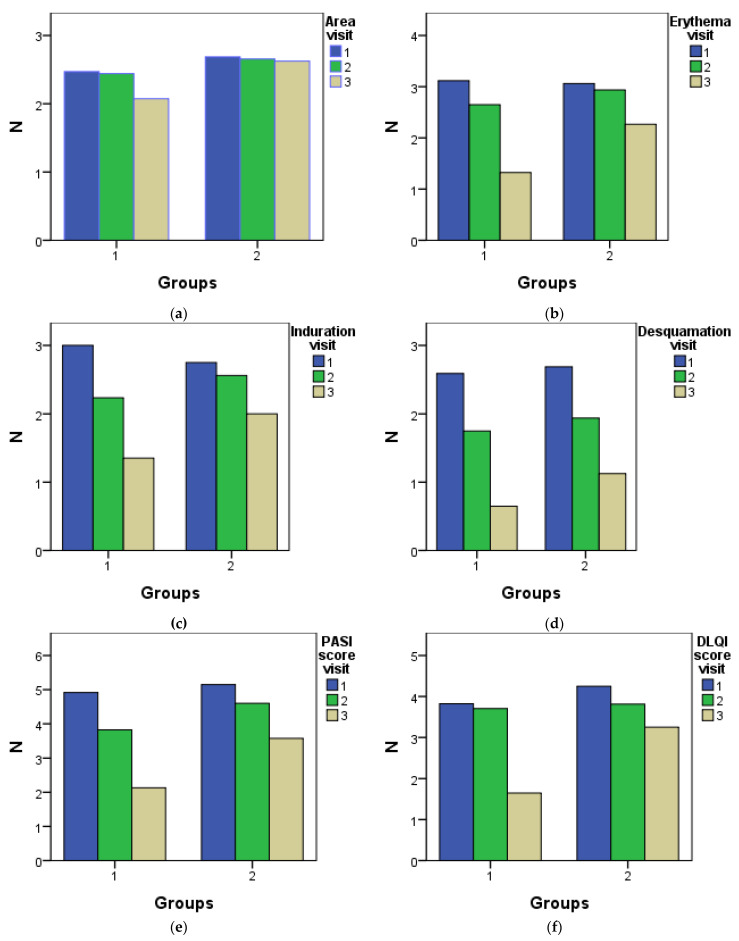
The parameters of the 2 groups compared during the 3 visits: (**a**) area; (**b**) erythema; (**c**) induration; (**d**) desquamation; (**e**) PASI score; and (**f**) DLQI score. Group 1 was treated with topical ointments containing *Rosa × damascena* Mill. extract, while Group 2 was used as a placebo group that received ointment with the same composition as that of the first group, except for the extract with *Rosa × damascena* Mill.

**Figure 2 pharmaceuticals-17-01092-f002:**
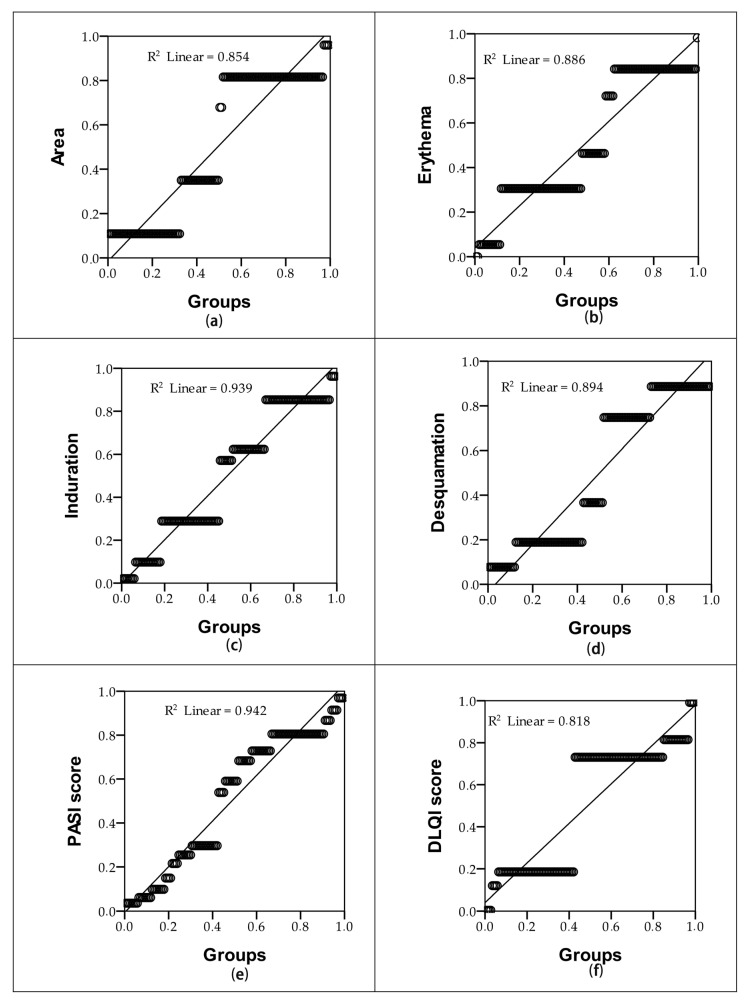
Evaluating the risk of an increase in the following parameters for the placebo group: (**a**) area; (**b**) erythema; (**c**) induration; (**d**) desquamation; (**e**) PASI score; and (**f**) DLQI score.

**Table 1 pharmaceuticals-17-01092-t001:** Comparative statistics of the parameters between the three visits for the two groups.

Parameters		N	Mean	Std. Deviation	F	Sig
Area_1	1.00	58	2.4706	0.50285	5.212	0.024
	2.00	54	2.6875	0.58757		
	Total	112	2.5758	0.55425		
Area_2	1.00	58	2.4412	0.50022	5.057	0.026
	2.00	54	2.6563	0.59678		
	Total	112	2.5455	0.55758		
Area_3	1.00	58	2.0735	0.88632	16.369	0.001
	2.00	54	2.6250	0.65465		
	Total	112	2.3409	0.82730		
Erythema_1	1.00	58	3.1176	0.87297	0.184	0.669
	2.00	54	3.0625	0.55990		
	Total	112	3.0909	0.73572		
Erythema_2	1.00	58	2.6471	0.76811	5.374	0.022
	2.00	54	2.9375	0.66368		
	Total	112	2.7879	0.73131		
Erythema_3	1.00	58	1.3235	1.01395	38.283	0.000
	2.00	54	2.2656	0.69561		
	Total	112	1.7803	0.99091		
Induration_1	1.00	58	3.0000	0.77267	3.525	0.063
	2.00	54	2.7500	0.75593		
	Total	112	2.8788	0.77193		
Induration_2	1.00	58	2.2353	0.88297	5.464	0.021
	2.00	54	2.5625	0.70991		
	Total	112	2.3939	0.81735		
Induration_3	1.00	58	1.3529	1.03325	20.502	0.001
	2.00	54	2.0000	0.50395		
	Total	112	1.6667	0.87951		
Desquamation_1	1.00	58	2.5882	0.60434	1.105	0.295
	2.00	54	2.6875	0.46718		
	Total	112	2.6364	0.54244		
Desquamation_2	1.00	58	1.7500	0.55651	4.636	0.033
	2.00	54	1.9375	0.43187		
	Total	112	1.8409	0.50689		
Desquamation_3	1.00	58	0.6471	0.68599	25.411	0.001
	2.00	54	1.1250	0.33333		
	Total	112	0.8788	0.59297		
PASI_score_1	1.00	58	4.9176	2.63620	0.330	0.567
	2.00	54	5.1500	1.93284		
	Total	112	5.0303	2.31615		
PASI_score_2	1.00	58	3.8235	2.35671	4.365	0.039
	2.00	54	4.6000	1.86803		
	Total	112	4.2000	2.16114		
PASI_score_3	1.00	58	2.1294	2.20928	17.376	0.001
	2.00	54	3.5750	1.72948		
	Total	112	2.8303	2.11205		
DLQI_score_1	1.00	58	3.8235	1.93121	2.178	0.142
	2.00	54	4.2500	1.30931		
	Total	112	4.0303	1.66664		
DLQI_score_2	1.00	58	3.7059	1.82077	0.152	0.697
	2.00	54	3.8125	1.24563		
	Total	112	3.7576	1.56352		
DLQI_score_3	1.00	58	1.6471	1.85894	33.218	0.001
	2.00	54	3.2500	1.25988		
	Total	112	2.4242	1.78253		

**Table 2 pharmaceuticals-17-01092-t002:** Regression table of the parameters according to the research groups.

Parameters	Correlate	R^2^ (%)	r	β1	β2	t	F
Area	Groups	86.4	0.421 **	0.521 **	0.859 **	4.640 **	9.199 **
Erythema		93.6	0.670 **	0.519 **	0.397 **	9.134 **	34.822 **
Induration		95.2	0.595 **	0.138 *	0.242 *	1.934 *	23.350 **
Desquamation		92.9	0.417 **	0.367 **	0.433 **	4.645 **	8.972 **
PASI score		97.6	0.647 **	0.295 **	1.241 **	5.077 **	30.668 **
DLQI score		95.6	0.827 **	0.529 **	1.869 **	15.020 **	92.627 **

R2 = R square; β1 = unstandardized coefficient; β2 = standardized coefficient; t = coefficient of correlation; F = ANOVA coefficient; ** correlation is significant at the 0.01 level (two-tailed); * correlation is significant at the 0.05 level (two-tailed).

## Data Availability

Data are available from the corresponding author with the permission of the head of the department. The data that support the findings of this study are available from the corresponding author (bacskay.ildiko@pharm.unideb.hu) upon reasonable request.

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
