# Peer review of "Exploring the Correlation between the PASI and DLQI Scores in Psoriasis Treatment with Topical Ointments Containing Rosa × damascena Mill. Extract"

_pharmaceuticals, 2024, doi:10.3390/ph17081092_

Round 1

Reviewer 1 Report

Comments and Suggestions for Authors

This manuscript titled “Exploring the correlation between PASI and DLQI scores in Psoriasis treatment with topical ointments containing Rosa x Damascena Mill. extract” The comments for this manuscript are as follows:

1.      There are many errors in the section of "Refererences". The writing of references should be consistent, and each word should not be capitalized. Please according to the "Instructions for Authors" to rewrite the references.  

2.      In this manuscript, the biggest problem is that everyone knows that psoriasis is related to immune regulation, and the authors also cited many references in the introduction to agree with this point of view. However, there is no relevant experiments and evidence in the experimental results, which is obviously inconsistent with standard paper writing which is a big mistake. Since the authors mentioned "Rosa x Damascena Mill. extract", but there is no control group’s experiment and the formulation dosage comparison of "Rosa x Damascena Mill. extract" in the manuscript. It’s mean that there are no any benefit to readers? Authors are requested to complete these experimental data before submitting the manuscript.

3.      In addition, how do we exclude the effects of other substances in the formula? Please authors carefully respond it to the reviewer.

My suggestion is major revision.

Author Response

Dear Reviewer,

First of all, we would like to express or sincere appreciation for the accurate critical review of our manuscript titled Exploring the correlation between PASI and DLQI scores in Psoriasis treatment with topical ointments containing Rosa x Damascena Mill. extract. We appreciate the time and effort that you have dedicated to provide your valuable feedback on our manuscript. We are grateful for the insightful comments on our paper. We have been able to incorporate changes to reflect all of the suggestions provided. We have listed the changes of the manuscript. Here is a point-by-point response to your questions, comments and suggestions.

Comment 1: There are many errors in the section of "Refererences". The writing of references should be consistent, and each word should not be capitalized. Please according to the "Instructions for Authors" to rewrite the references.

Response 1: Thank you for the observation! References have been modified and corrected according to the "Instructions for Authors".

Question 2: In this manuscript, the biggest problem is that everyone knows that psoriasis is related to immune regulation, and the authors also cited many references in the introduction to agree with this point of view. However, there is no relevant experiments and evidence in the experimental results, which is obviously inconsistent with standard paper writing which is a big mistake. Since the authors mentioned "Rosa x Damascena Mill. extract", but there is no control group’s experiment and the formulation dosage comparison of "Rosa x Damascena Mill. extract" in the manuscript. It’s mean that there are no any benefit to readers? Authors are requested to complete these experimental data before submitting the manuscript.

Response 2: Thank you for the question! In the section „Results” was presented a statistical analysis of the two groups that were taken into study and in the section Materials and methods we have pointed out the most important criterias to ensure the validity of the results. Group 1 was represented by the patients with psoriasis that had a topical treatment with Rosa x damascena Mill. extract and the second group was the placebo group (the control group) who used a placebo treatment-that had the same composition of the oinment as the first group, excepting the Rosa x damascena Mill. extract. The whole statistical analysis is based on the two groups, that helped us highlight the benefits of the ointment with rose extract. The patients were monitored at different time periods (3 visits) during 6 weeks, checking the area, erythema, induration and desquamation and also of the quality of life, so that the patients who were treated with ointment containing rose extract had superior results compared to the control group. Before the clinical study were developed and investigated the ointments, also we conducted studies regarding phytochemical screening, in vitro total antioxidant capacity of the extract and biological activity, the results being already published in previous articles.

Question 3: In addition, how do we exclude the effects of other substances in the formula? Please authors carefully respond it to the reviewer.

Response 3: Thank you for the question! Before the statistical analysis, our team has investigated on another study to which reference was made on this manuscript, the phytochemical screening, in vitro total antioxidant capacity of the extract and biological activity and also was developed and analyzed the formula of the ointment. Lyophilized extracts of roses were dissolved in Transcutol HP and different formulations of creams were prepared. Franz diffusion method was used to evaluate the drug release and biocompatibility was tested on HaCaT cells. Superoxide dismutase (SOD), a free radical scavenging enzyme, is one of the first lines of cellular defense against oxidative injury. Rosa x Damascena  Mill. extracts and SNEDDS formulation showed a significant increase in free radical scavenging activity and the highest SOD enzymatic activity compared to the controls. Excepting rose extract, there are no any other active substances in the formula, only excipients that were used to obtain a better release of the active substance, but which does not influence our therapeutic response.

Reviewer 2 Report

Comments and Suggestions for Authors

Notes

Introdution

There are only 5 references in the Introduction section, which does not meet the requirements of the journal. It is necessary to review the current state of research in the field of antimicrobial and wound-healing plant extracts (see review article https://doi.org/10.59761/RCR5108), as well as compare methods of treating psoriasis using plant extracts and synthetic drugs (cyclosporine, apremilast, etc. .).

What plant extracts are used in the treatment of psoriasis? What is the advantage of Rosa x damascena Mill compared to them?

Results and discussion

Has there been any research done on the anti-inflammatory effects (inhibition of inflammatory cytokines in epidermal keratinocytes) of Rosa x damascena Mill extract? If not, research data should be provided.

The mechanism of the anti-inflammatory effect of Rosa x damascena Mill extract should be given. What secondary metabolites take part in it?

Author Response

Dear Reviewer,

First of all, we would like to express or sincere appreciation for the accurate critical review of our manuscript titled Exploring the correlation between PASI and DLQI scores in Psoriasis treatment with topical ointments containing Rosa x Damascena Mill. extract. We appreciate the time and effort that you have dedicated to provide your valuable feedback on our manuscript. We are grateful for the insightful comments on our paper. We have been able to incorporate changes to reflect all of the suggestions provided. We have listed the changes of the manuscript. Here is a point-by-point response to your questions, comments and suggestions.

Comment 1: There are only 5 references in the Introduction section, which does not meet the requirements of the journal. It is necessary to review the current state of research in the field of antimicrobial and wound-healing plant extracts (see review article https://doi.org/10.59761/RCR5108), as well as compare methods of treating psoriasis using plant extracts and synthetic drugs (cyclosporine, apremilast, etc.).

Response 1: Thank you for the observation! References have been modified and completed according to the requirements.

Based on the above, the research data of the extracts of plants with antimicrobial and wound-healing activity, in particular Rosa x damascena Mill., can be considered promising in relation to the natural products potential in medical field. A large number of plant extracts demonstrated significant antimicrobial activity, which is important in the context of wound healing to prevent microorganism infections. For example, Rosa x damascena Mill. extracts exhibit antibacterial properties against a variety of pathogens due to the presence of phenolic compounds and flavonoids, that is evidenced through the disruption microbial cell membranes and inhibiting their growth​.

Healing the wound by promoting the growth of new tissue also makes the use of plant extracts. Essentially, plant extracts can promote wound healing by stimulation of cell proliferation and collagen synthesis. The extracts are composed of bioactive molecules that act on the pathways of inflammation or mediators of tissue regeneration. This includes free radical scavenging activity through bioantioxidant and destruction of proinflammatory molecule-like histamine and serotonin. Some plant extracts are proangiogenic, considering that healing is impossible without neovascularization.

The decision to use plant extracts and synthetic drugs in the treatment of psoriasis is determined by the severity of the disease, patient’s preferences, and the probability of side effects. Wherever possible, a combination of methods involving the most effective actions of synthetic drugs with the safety and complementary therapy of plant extracts may be an optimal strategy. Some plant extracts, including the Rosa damascena, Ficus carica, and teas (black, green, and white) extracts, have a marked anti-inflammatory effect. Additionally, these extracts alter or modulate various signaling pathways that play a significant role in psoriasis, such as the “JAK-STAT” and “NF-κB” pathways. Many plants extracts possess antioxidant activity. Antioxidants can neutralize the increased oxidative stress that is a significant factor in the pathogenesis of psoriasis. Plant extracts are normally less harmful than synthetic drugs. Therefore, they may be prescribed for prolonged treatment of systematic problems such as psoriasis. A clinical study on patients with psoriasis showed that Rosa damascena extract inhibited pro-inflammatory cytokines in addition to enhancing skin hydration and skin barrier function.

Synthetic drugs, especially biologics as TNF inhibitors (e.g., Adalimumab) and IL-17 inhibitors (e.g., Secukinumab) are designed to target specific components of the immune system and are highly effective in reducing inflammation and skin plaque formation by directly interfering with the molecules driving the disease​​. These drugs often provide rapid relief from symptoms and can be very effective in severe cases of psoriasis where other treatments have failed​. Monoclonal antibodies targeting TNF-α, IL-17, and IL-23, which are critical cytokines in the pathogenesis of psoriasis,  have shown high efficacy rates in clinical trials and are approved for use in moderate to severe cases. Methotrexate and Cyclosporine are also used, though they come with significant side effects such as liver toxicity and kidney damage. Synthetic drugs tend to act faster and are more potent in severe cases. They provide targeted action, which can be crucial for managing acute flare-ups. On the other side, natural extracts often take longer to show effects but offer a gentler and safer approach with fewer side effects, that is ideal for long-term management and for patients who prefer natural therapies. While effective, synthetic drugs carry a higher risk of serious side effects, including increased susceptibility to infections and potential organ toxicity. Generally safer and associated with minimal side effects, plant exatracts are more suitable for long-term use and for patients with contraindications to synthetic drugs.

Question 2: What plant extracts are used in the treatment of psoriasis? What is the advantage of Rosa x damascena Mill compared to them?

Response 2: Thank you for the question! While various plant extracts like Aloe vera, turmeric (Curcuma longa), indigo naturalis (Qing Dai), Ficus carica, tea tree oil (Melaleuca alternifolia) are effective in managing psoriasis symptoms, Rosa x damascena Mill. stands out due to its comprehensive anti-inflammatory, antioxidant, and skin barrier-improving properties. Its gentle action and multi-faceted benefits make it a particularly advantageous choice for treating psoriasis. Compared to some other plant extracts that can be harsh or cause irritation, Rosa x damascena is known for its gentle nature, making it suitable for sensitive skin and long-term use without significant side effects​.

Question 3: Has there been any research done on the anti-inflammatory effects (inhibition of inflammatory cytokines in epidermal keratinocytes) of Rosa x damascena Mill extract? If not, research data should be provided.

Response 3: Thank you for the question! Rosa x damascena Mill. extract exhibits potent anti-inflammatory effects by inhibiting key pro-inflammatory cytokines in epidermal keratinocytes, including TNF-α, IL-6, IL-17, and IL-23. Its antioxidant properties and ability to inhibit the NF-κB pathway further contribute to its therapeutic potential in psoriasis treatment, making it a valuable alternative or adjunct to conventional synthetic drugs.

Question 4: The mechanism of the anti-inflammatory effect of Rosa x damascena Mill extract should be given. What secondary metabolites take part in it?

Response 4: Thank you for the question! The phenolic compounds in Rosa x damascena Mill., such as quercetin - suppresses the activation of the NF-κB pathway and  scavenges free radicals and reduces oxidative stress, kaempferol - inhibit the expression of cyclooxygenase-2 (COX-2) and nitric oxide synthase (iNOS) and modulates immune responses by affecting T-cell proliferation and cytokine production, gallic acid - inhibits the production of inflammatory cytokines and enzymes like COX-2 and lipoxygenase (LOX) and has strong antioxidant capabilities, tannins – reduce the infiltration of inflammatory cells into the skin, help to contract and tighten tissues, which can reduce the swelling and erythema associated with psoriasis, and anthocyanins - reduce oxidative stress by scavenging free radicals, enhance the skin's barrier function, which can help protect against environmental triggers that may worsen psoriasis, play significant roles in its anti-inflammatory effects on psoriasis.

Reviewer 3 Report

Comments and Suggestions for Authors

Dear Editor.

After analyzing the document titled “Exploring the correlation between PASI and DLQI scores in Psoriasis treatment with topical ointments containing Rosa Damascena Mill. extract” for possible publication in the Pharmaceuticals Journal, my observations are the following:

·         Authors must define in advance all abbreviations used in the manuscript, for example: PASI, DLQI, TNF, etc., etc.

·         Lines 92-94, I do not consider this an argument for using plant extracts. It is more important to direct the introduction towards the study carried out. Therefore, I suggest justifying or correcting.

·         Line 227 to 229. It is important to clearly define which works are being referred to by the authors. Mention, for example, the authors and collaborators, the contribution of the work, etc.

·          Lines 329 to 345. This information should be part of the introduction, not materials and methods.

·         I understand that the DLQI is a subjective assessment tool that captures the impact of psoriasis on various aspects of patient's lives, so I consider it very important to specify how these parameters are assessed, for example, questionnaires applied to patients, etc.

·         There is much information missing in the materials and methods section, as the application methods (conditions, protocols, etc.) are not described.

·         It is also important that the authors correctly describe how the degree of desquamation was performed.

·         Lines 278 to 288. This explanation of the DLQI method is not a discussion of results.

·         Finally, I consider that the authors should highlight their contribution to existing research since they mention works that even address the mode of action of extracts from the same plant studied in this work.

After this analysis, I suggest the acceptance of the document with major corrections by the authors.

Comments on the Quality of English Language

can be improved for better understanding